# Developing a Cybersecurity Framework for e-Government Project in the Kingdom of Saudi Arabia

**Abdullah Alrubaiq *** and **Talal Alharbi ***

Department of Information Technology, College of Computer and Information Sciences, Majmaah University, Al Majmaah 11952, Saudi Arabia
* Correspondence: 411103746@s.mu.edu.sa (A.A.); talal@mu.edu.sa (T.A.)

**Abstract:** The evolution of information systems has escalated significantly within the last decade as research unveils new concepts. The general orientation to provide solutions to complex problems continues to inspire innovation and new advancements. Cybersecurity is emerging as a critical factor for consideration within the resultant paradigm as information systems become significantly integrated. This paper provides an in-depth analysis of cybersecurity within the context of information systems. The paper examines some of the most consequential aspects of cybersecurity from the perspective of an e-government project in Saudi Arabia. A holistic system is proposed within the research framework to incorporate various scientific guidelines. The general orientation of this research is predicated on the aspiration to design and implement a complex and robust framework within which an e-government system can thrive within the Saudi Arabian context. A consideration of the physical environment within which the project will operate is also made, focusing on security. An evaluation of the cybersecurity environment in Saudi Arabia is reflective of significant advancements that have occurred in information system domains within the past few years.

**Keywords:** cybersecurity; information systems; information system infrastructure; e-government

## 1. Introduction

### 1.1. Background

Government represents a diversity of functions, structures and objectives. The government machinery of different countries operates on a dynamic system of conflicting and complementary variables [1]. The implementation of e-government systems occurs within the complexity of such systems and requires extensive considerations around security [2]. The use of technology and the Internet to process transactions for public benefit continues to evolve over time [3]. The primary objective of implementing e-government systems for most countries is to enhance service delivery and align the technological evolution happening in the contemporary world [4]. Various variables, including accountability, transparency, and citizen participation, accompany the implementation of e-government systems [5]. The implementation of e-government systems is a comprehensive affair that requires the careful consideration of various security variables within cyberspace. There have been various advancements in the operationalization of the concept from its simple origins, with various countries adopting integrated systems to improve the way they deliver services to their citizens [6]. Citizens of various countries can access a diversity of services through automated and inter-linked systems [7]. Some e-government systems are predicated on technological frameworks that can collect, process and interpret information [8]. The use of e-government systems is essential as it improves delivery of services to citizens, businesses, and among government agencies [9]. The security of future e-government systems is indispensable, in this regard, because of the benefits that accrue from the operationalization of such systems. This paper provides a comprehensive outlook on the functioning of e-government in Saudi Arabia to investigate security measures.

Cybersecurity remains a significant area of interest within e-government operations in Saudi Arabia.

*1.2. Problem Evolution*

There are various issues that necessitate an investigation on the importance of cybersecurity in e-government systems such as that of Saudi Arabia. The e-government model of government systems is a relatively modern conceptualization with its origins in the 1980s [1]. The implementation of technology in delivering diversified services within a government context is a costly affair that requires safeguarding for economic prudence [10]. The basic conceptualization of e-government systems is creating a robust framework within which citizens can access a plethora of services and must be protected to secure social welfare [11]. The implementation of e-government in Saudi Arabia has induced significant benefits, including the encouragement of digital technologies that are important to economic competitiveness, re-orientation of government systems to becoming citizen-focused and a reduction in the costs of public service [12]. The case of Saudi Arabia presents other dynamics, such as the competitiveness of the government in achieving efficiency and maintaining a positive outlook on the global stage for securing diverse economic interests.

Governments such as that of Saudi Arabia have recognized the importance of building complex e-government structures as a pathway to socioeconomic development. Such systems can be improved in terms of accelerating service delivery to customers if they are encapsulated within a robust cybersecurity environment. There is a necessity for global governments to invest in cybersecurity infrastructure and systems to enhance the manner in which they provide diverse services to their citizens [13]. The creation of robust e-government systems is also critical for the maintenance of a steady progression toward technological advancements in line with other global powers [14]. Cyber threats pose significant risks to the implementation of e-government systems. To oversee the elimination of the threats, associations ought to comprehend the probability that an occasion will happen and the potential coming about the effects. An investment in the cybersecurity framework of e-government systems is a basic necessity for the future of global governments and the collective development of the world socioeconomic fabric.

Formulating the research problem within the constructs of this paper is achieved by systematically identifying and improving the design constructs, which can be defined as the language in which the problem and solution are defined. Design constructs in the context of this paper are the factors that affect the success of e-government implementations. A comprehensive review of the body of literature on e-government and cybersecurity in Section 2 indicates that there are various challenges incidental to the success of implementing e-government systems within any environment. The literature further shows that the implementation of e-government systems provides for a mixture of public service, software development, and systems integration. The different constructs of this research are not created from nothing but arise as a consequence of the extracted success factors from a literature review. The success factors have been reviewed and refined in Section 2 to avoid duplication.

*1.3. Key Definitions*

1.3.1. Cybersecurity

Cybersecurity refers to the process of protecting and recovering computer systems, programs, networks, and devices from cyber-attacks. There are numerous cyber threats that are incidental to e-government systems, such as that of Saudi Arabia, that are yet to be identified due to the sophistication of such issues [15]. The importance of cybersecurity on e-government platforms is enormous, since the systems contain personal data, for instance, that could result in identity theft, among other issues [16]. The e-government systems of Saudi Arabia are susceptible to cyber-attacks from adversaries and criminals with different intentions and have to be protected from a multi-stakeholder and multi-dimension perspective [8]. Cyber attackers employ new methods driven by social engineering and artificial

intelligence in circumventing traditional security systems [17]. Cyber-attacks have increasingly become sophisticated, posing danger to sensitive data in contemporary society.

### 1.3.2. Information Systems

Information systems play a pivotal role in interpersonal and inter-organizational interactions within the contemporary world. They are sociotechnical, formal, and organizational systems designed to gather, process, store, and distribute information on a large scale [12]. They comprise four basic components: technology, structure, people, and tasks. They entail an integrated collection of components that allow for the gathering, storage, and processing of data that is consequently used to contribute to bodies of knowledge and provide information within digital circles [1]. A computer information system is used to collect and interpret information in different contexts [18]. Information systems support operations, management, and decision making by using computer systems. Cybersecurity systems must be integrated into any information system to ensure that their general integrity and functionality is safeguarded.

### 1.3.3. e-Government

e-government is the short form for electronic government and refers to the framework of using information systems and technological communications to provide services to citizens within government. e-government encompasses a diversity of variables, including the use of the Internet and devices such as computers to simplify the work of government in providing various services to citizens [1]. The adoption of an e-government model can provide numerous opportunities for different stakeholders and enhance the convenience and efficiency of services [18]. Citizens can access government services directly without going through different intermediaries within a properly functioning e-government system [12]. It becomes incumbent for government to ensure that such systems are protected, and therefore, to ensure that the entire paradigm of providing services to people is secured. Cybersecurity is an integral aspect of any e-government framework.

### *1.4. Research Objectives*

The general objective of this research is to establish the level of cybersecurity within the e-government framework of Saudi Arabia. The paper examines the extent to which the entire infrastructure of information systems used to provide a plethora of services to the citizens of Saudi Arabia is secure. An in-depth analysis of the various variables around the cybersecurity framework of Saudi e-government services is performed to identify areas of weakness. A general appraisal of the cybersecurity framework in the e-government environment of Saudi Arabia is contextualized with a comprehensive comparison with other countries of similar status.

Specific Objectives

The following specific objectives are considered for this research's:
- To identify areas of weakness in the cybersecurity framework in the e-government operations of Saudi Arabia.
- To establish the level of strength in the cybersecurity framework of the e-government operations of Saudi Arabia.
- To compare the strength of cybersecurity framework of the e-government of Saudi Arabia and other countries of the world.

### *1.5. Research Questions*

The following research questions have been formulated to achieve the specific objectives of this research:
- What are the areas of weakness in the cybersecurity framework of the e-government mechanisms of Saudi Arabia?

- What are the strengths of the cybersecurity framework in the e-government operations of Saudi Arabia?
- What are the similarities and differences between the cybersecurity framework of the e-government of Saudi Arabia and other countries of the world?

*1.6. Structure of Paper*

This paper begins with a comprehensive introduction that lays out the research general orientation. The introductory part of the paper begins with a general background and evolution of the problem. An overview of the entire paper is presented to provide a proper understanding of the subject of research. These parts provide a general context of the research and predicate the basis for outcomes in subsequent sections. Key definitions of the various terms used within the context of research are provided within the introduction of the paper as well. The introductory part concludes with the research objectives and questions used within the research framework.

The remainder of this paper is organized as follows: Section 2 discusses the related works, and Section 3 presents the methodology used to assess the cybersecurity awareness level. Section 4 describes the analysis results based on the dataset collected in this study. Section 5 presents the findings of the statistical tests performed in this study, and Section 6 concludes the paper.

*1.7. Conclusions*

In conclusion, e-government systems across the world face numerous threats that emanate from the cyberspace in which they operate. The case of Saudi Arabia provides a suitable basis for investigating such threats to prescribe effective solutions. Saudi Arabia is developing an increasingly complex e-government infrastructure that needs to be protected to effectively deliver services to citizens [12]. There is also a necessity to examine some threats that exist within the macroenvironment to make predictions about future threats. A comparison between the cybersecurity framework of Saudi Arabia and that of other countries is necessary to assess the necessity for improvements. This paper provides a robust foundation for evaluating the cybersecurity environment within e-government operations in Saudi Arabia.

## 2. Literature Review

Cybersecurity regarding e-governments is an essential topic for consideration due to the importance of effective service delivery for societal growth and development. The government, with its citizenry, derives significant benefits from the use of information systems to conduct various activities [1]. Dias put significant issues into perspective in the analysis of the significance of e-government in broad economies. The purpose of the study conducted by Dias is to research whether it is possible to determine the relevance of policies around e-government by discounting the impact of relative wealth for countries. Implementation of e-government systems requires an elaborate and robust cybersecurity framework from an analysis of this and other studies as will be observed in the literature review. Cybersecurity has been observed to be among the most significant issues of national importance. Cybersecurity not only protects information from being stolen but also safeguards the safety and interests of various users [10]. Cyber security is an essential element of national security and safekeeping of assets according to Talib et al. (2018). Talib and other researchers further note that Saudi Arabia is a prime target for cyberattacks among all Arab countries. The paper was particularly interested in the use of ontology to identify and suggest encoded and forma descriptions of cybersecurity strategies with a focus on contemporary studies done in Saudi Arabia. There are numerous possibilities for information getting into the wrong hands and being used to perpetrate crime and other malicious activities. Protection of the integrity of government information is also essential to inspire confidence among citizens. Cybersecurity regarding the operations of e-governments can be quite comprehensive and must be approached from a multifaceted perspective.

## 2.1. Related Studies

The case of Saudi Arabia provides the complexity required to examine the importance of cybersecurity within a diversified environment. Saudi Arabia has emerged as one of the most competitive economies of the 21st Century with significant economic strength on the global socioeconomic landscape [19]. The research conducted a survey on the perceptions of the people of Saud Arabia regarding characteristics of e-governments. Aljarallah and Lock observe that sustainable e-government operations have become an essential consideration for different governments. The paper further notes that there is a buss around including e-government within different political systems. Aljarallah and Lock note that there is sparse literature on sustainability in e-governments; however, necessitating quantitative empirical studies in this area. Innovation is one of the most essential yardsticks of the growth that has characterized countries of the Middle East region over the past few years. Cybersecurity presents itself as a consequential factor if issues such as government operations are considered. The Saudi Arabian government is increasingly investing in technological innovations in the fields of finance, commerce, social welfare, health, security and defense [20]. Securing cyber spaces can no longer be viewed as a consideration but as a necessity, in this context, as the government aspires to influence positive outcomes. The Saudi Arabian government continues to invest in significant efforts to improve the cybersecurity environment within which it seeks to facilitate service delivery. Albrahim and other researchers noted that e-government services require considerable interconnection among agencies and intensive exchange of information to provide specialized online services. The paper further noted that such specialized online services would then in turn be used to improve decision making. Cybersecurity issues would compromise the availability, confidentiality and integrity of information being exchanged across different government networks from the perspective of the research. The paper further noted that it was the responsibility of government to guarantee the protection of information within an e-government system. The paper noted that most of the e-government frameworks adopted by various governments are still weak and would need advanced modelling for optimized outcomes. Government agencies in Saudi Arabia have adopted various communication and information technologies to improve the way they serve the general populace. New technologies have enhanced not only the delivery of essential services but also the outlook on the technological direction of Saudi Arabia looking into the future [21]. There is a necessity to protect these paradigms as the government seeks to compete favorably for tourism in business, education, and tourism in hospitality and medicine [22]. Various processes are predicated on e-business and e-government platforms in providing services to citizens. Almukhlifi and other researchers presented an investigation of the moderation effect of the Saudi Arabian culture that is known as Wastta on the integration of e-government in Saud Arabia. This investigation takes the perspective of the general citizens of the country in evaluation the impact of e-government on society and the necessity to protect such systems. A hierarchical multiple regression analysis is conducted within the context of the research on the data gathered through a survey in the country. The study indicates that Wastta influences the adoption of e-government in Saudi Arabia because of the moderation effect on the perceived simplicity and utility of e-government. This research brings into focus the importance of interrogating the influence of culture on e-government systems. Alassim and other researchers investigated the organization variables incidental on the operationalization of e-government systems within the public sector in Saudi Arabia. There are various factors that can influence the effective implementation of e-government systems, and the adjoining cybersecurity considerations, from an organizational point of view. The Yesser program was established in 2005 with the view of controlling the transformation process of e-government systems in Saudi Arabia. The objectives of the project were to provide an environment of collaboration and for government entities to implement e-government systems to improve efficiency and effectiveness within the public sector. Considering organizational challenges in the implementation of e-government systems and the consequent cybersecurity is critical in improving service delivery in the

long run. Solutions to the problem of cybersecurity within e-government machinery of Saudi Arabia requires the systematic development of the framework core. The framework core comprises of a set of cybersecurity activities and expected outcomes whose application and references are common across critical infrastructural sectors [17]. The core defines standards applicable in the industry as well as the practices and guidelines that need to be promoted to create a secure environment within government operations. Cybersecurity at any level of government can be optimized through systematic implementation across different agencies. The core framework is comprised of five parallel and continuous aspects that include the identification, protection, detection, response and recovery of essential data [18]. Considerations around the functions of the cybersecurity framework can give a high level and strategic focus of the lifecycle of e-government management of cybersecurity related risks. The framework core can be said to be fundamental in ascertaining the key discrete outcomes for each function of the e-government platform. The framework implementation tiers contextualize the relationship between e-government systems and cybersecurity risks. This contextualization takes place in a comprehensive framework with robust considerations of how to mitigate risks. The tiers are used to provide a description of the extent to which an organization and its cybersecurity risk reduction management methodologies are illustrative of the characteristics defined within the framework [17]. The tiers try to depict the practices of an organization over a range from partial in the first tier to adaptive in the fourth tier. The different tiers reflect the progression of an organization from informal, reactive responses to more flexible and agile risk-informed approaches. Organizations must consider their current organizational constraints, business objectives, legal and regulatory requirements, threat environment and risk management practices [19]. A consideration of the extent to which an environment is secure is a primary consideration in the determination of effectiveness. There is sufficient evidence to the effect that a majority of the citizenry of Saudi Arabia are not technical people and do not have the capacity to respond to cybersecurity issues as and when they arise. This postulation speaks to the necessity of a country to be self-sufficient in terms of technological advancement and not rely on external parties for such growth. The e-government niche might be a very minute section of government operations but is quite consequential in terms of service delivery [17]. The government of Saudi Arabia must ensure that it remains cognizant of the challenges that lay ahead and build capacity among its own people [12]. There is eminence of a dispensation where technology will be a frontier in the competition of global powers. Saudi Arabia must ensure that such a time will not only find it prepared at the macro level but also at the micro level. The services delivery system of Saudi Arabia might be at a significant cybersecurity risk to external forces with inadequacy of capacity among local personnel. There is significant uncertainty about the requirement for re-engineering business measures in the e-government systems of Saudi Arabia. Business measure re-engineering or overhauls should be predicated on the necessity for effective execution of e-government operations in Saudi Arabia [17]. Execution of e-government ventures within the context of Saudi Arabia needs significant re-evaluation and audits of previous cycles. Privacy and security issues are frequently cited within research but are yet to be fully understood within the context of Saudi Arabia [16]. The technical capacity of the various stakeholders in the context of Saudi Arabia can also be put into question given the countries reliance on external parties for advisory and technical expertise. Threats from intruders and hackers have not been adequately mitigated in the case of Saudi Arabia necessitating a more complex and robust approach towards issues of cybersecurity. Different associations can decide the satisfactory level of danger for accomplishing their hierarchical destinations and can communicate this as their danger resistance from a review of literature. associations can organize online protection exercises, empowering associations to settle on educated choices about network protection consumptions with a proper contextualization and understanding of resilience [19]. The systematic usage of danger board programs offers associations the capacity to evaluate and adopt changes in accordance with their online protection programs. Different associations may decide to deal with risk through different

other approaches including tolerating danger, avoiding danger, moderating danger and moving it to other areas [12]. The e-government system of Saudi Arabia seems to utilize cyber threats elimination methods to push the executive cycles to empower associations to organize and advise choices with respect to the safety of the entire network [16]. This approach underpins a repetition of the hazard evaluation and approval of business drivers to help different associations with selecting objective states for network safety. These approaches can significantly help to improve the ability to achieve desired outcomes. The network protection industry can profit by blockchain's exceptional highlights, which make a basically invulnerable divider between a programmer and your data. The straightforward record considers secret phrase free section. Utilizing biometrics, including retina sweeps and fingerprints, the record can make a solitary source, uncrackable type of passage into any private information. Decentralized capacity guarantees that each square contains just a little enlightening piece to a lot bigger riddle, restricting hackable information to barely anything. At last, blockchain's freely available report keeping framework gives every hub a knowledge into any information control, uncovering potential digital wrongdoing endeavors continuously. Blockchain in network safety is far reaching, and we have gathered together six enterprises that utilization it as another weapon in the battle to ensure our most delicate data. First carried out as the operational organization behind Bitcoin, blockchain is currently utilized in excess of 1000 distinctive cryptographic forms of money, a number that becomes practically every day. DLT secures the trustworthiness of cryptos through encryption techniques and public data sharing. The authenticity of digital money buys by people is guaranteed in light of the fact that they can follow the exchange of the cash to its starting point. Encryption helps control the measure of digital currencies being made, subsequently settling esteem. These four organizations use blockchain as a network safety convention in digital money exchanging.

No digital safeguard or data framework can be viewed as 100% secure. What is considered safe today will not be tomorrow given the rewarding idea of cybercrime and the criminal's inventiveness to look for new strategies for assault. Blockchain is acquiring foothold today, however pundits who question the versatility, security, and supportability of the innovation remain. Albeit some of blockchains basic abilities give information privacy, honesty and accessibility, very much like different frameworks, network safety controls and guidelines should be received for associations utilizing blockchains to shield their associations from outside assaults. The Cyber Physical System (CPS), Internet of Things (IoT) and Digital Twin are altogether focal ideas in Industry 4.0, regularly utilized reciprocally in conversations about Industry 4.0 and brilliant assembling. Each alludes to a portrayal of a piece of gear in the internet. Such portrayals are of focal significance in Industry 4.0 and for keen assembling, since they give admittance to ongoing operational information of the addressed hardware [23]. Utilization of this information goes from machine operational status and assembling significant KPIs, as OEE, MTBF, MTBA and so forth, to large information investigation and AI applications, like prescient upkeep. It is along these lines advantageous to look at what each means and how they identify with one another. The expression 'Digital Physical System' is said to have been authored without precedent for 2006 by Helen Gill of the National Science Foundation (NSF). The beginning of the expression 'Web of Things' is for the most part attributed to Kevin Ashton while at MIT in 1999, though the beginning of the expression 'Advanced Twin' is by and large credited to Michael Grieves while at University of Michigan in 2001 [24]. The expression 'Modern Internet of Things' was a new expansion to demonstrate the utilization of IoT in mechanical applications rather than purchaser applications.

### 2.2. Research Objectives

This literature review feeds into a comprehensive framework that meets the objectives of the paper. The literature review provides context into strengths and weaknesses of the cybersecurity framework of the e-government system of Saudi Arabia. The literature review further speaks to the requirements of future frameworks of e-government of Saudi Arabia

with a key focus on strengthening existing cybersecurity measures. There is a necessity for the government of Saudi Arabia to continue implementing sound approaches for the strengthening of cybersecurity within e-government operations as it competes with other players in the market [1]. The government of Saudi Arabia needs to also build local capacity in its bid to achieve effective delivery of service to its citizenry [18]. The literature review thus meets the expectations of the objectives of this paper and allows for an interrogation of future capabilities.

### 2.3. Gaps in Literature

There is a necessity to conduct further research on the sustainability of e-governments. Initiatives towards enhancing cybersecurity in e-government systems could be quite costly if such governments do not have sufficient monies to sustain such programs. Saudi Arabia might also be under the illusion that such systems are not expensive to run given its vast wealth emanating from oil fields. A further scrutiny of the true state of affairs is necessary with the view of making e-government systems sustainable. Different countries of the world must continually review their sustainability framework in terms of e-governments to ensure that they are able to meet the obligations around possible security challenges that may arise. Sustainability of cybersecurity systems in governments is as important as sustainability of e-government and government systems. The influence of culture must be taken into consideration in interrogating the importance of cybersecurity in e-government systems in the case of Saudi Arabia. There are various cultural influences incidental on the use of e-government systems in Saudi Arabia. It is essential to understand the influence of different aspects of culture on the acceptability of e-government systems, their utility, and overall attitude towards the adoption of cybersecurity. It is essential to consider the fact that there is a possibility for various internal attacks on the functioning of e-government systems if such systems are not welcome within a given context. A re-evaluation of the conservative nature and moderation within the Saudi culture is essential in the foregoing discussion on cybersecurity in e-government systems.

The literature review on e-government shows that there are various snags and achievement factors identified with e-government usage ventures within the country. The review further includes various viewpoints on the implementation of cyber security framework. Development thresholds and general operationalization of e-government systems in Saudi Arabia seems to be identical to those in created in other nations from a review of literature [10]. Saudi Arabia needs information and experience to deal with high innovation ventures since the country is not a primarily industrial or mechanical nation. The requirement for utilizing innovation and training is very useful in this regard and must be pursued from a multi-facet point of view [16]. The implementation of e-government across different sectors of continues to be a transformative factor that allows for the delivery of services in a transparent manner. The e-government system of Saudi Arabia is facing significant threats to cyber security leading to the requirement for a comprehensive cybersecurity framework.

### 3. Research Methodology

This research took a mixed qualitative and quantitative approach towards the investigation of the cybersecurity framework of e-government operations in Saudi Arabia. A review of relevant literature has been conducted to provide a basis for the discussion and analysis section of this paper. The research methodology deployed within this research framework has been optimized to thoroughly scrutinize various variables within the cybersecurity framework of the country. The results of the research were further validated by a comprehensive research survey that was conducted within government agencies and technology companies in Saudi Arabia. The research was modelled around the fact that the cybersecurity environment in the context of Saudi Arabia is significantly interconnected and interfaced with different stakeholders in technology. The research design adopted for this research was balanced targeted at the identification of strengths and weaknesses in the cybersecurity environment of the e-government framework in Saudi Arabia. A pool of

respondents was obtained to provide insight on the various aspects of the cybersecurity within e-government operations of Saudi Arabia. Perspectives regarding the cybersecurity framework of e-government collected through a survey administered by a questionnaire. The information collected was analyzed and inferences made through various statistical tools and interpreted for the drawing of conclusions and recommendations. The research was conducted within the robust constructs of ethics in research with all the interviewees being requested to provide their consent for participation. The respondents were also educated on the purpose of the research and the resultant implication of their participation in the long run. The research methodology that was adopted in this research exercise was optimized and balanced for generalization to large populations.

### 3.1. Methods of Data Collection

Various methods were used to collect the data that informed the current research on the extent of cybersecurity capabilities of the e-government machinery of Saudi Arabia. A comprehensive survey questionnaire was taken to collect relevant information for the research study on the extent and nature of cybersecurity strengths and weaknesses within the case of Saudi Arabia. The survey was also used to collect relevant information on a broad range of issues including the validity and practicality proposals for the improvement of cybersecurity in the e-government environment of Saudi Arabia. Data collected from the survey was placed against a body of information from the literature review. Peer reviewed literature was analyzed with the objective of examining the merits and demerits of the existing cybersecurity infrastructure. The data that was taken was analyzed by both primary and secondary researchers before being aligned and incorporated into the research framework.

### 3.2. Research Population—Respondents

The population of the research study exercise included persons who had the capacity to evaluate the cybersecurity environment in the e-government framework of Saudi Arabia and who understood general technical and non-technical concepts in the field of technology and cybersecurity in Saudi Arabia. They could make interpretations regarding the general security environment within the e-government and cyberspace incidental to government operations and were also able to make broad inferences regarding the future of cybersecurity in Saudi Arabia. The population selected for this research study consisted mainly of persons who work within the government of Saudi Arabia or who have had interactions with the government of Saudi Arabia. The population also comprised of persons with extensive experience in the field of information technology.

### 3.3. Research Tools

The primary tool of research in this study was a questionnaire with different sets of questions relevant to the topic of information technology and cybersecurity in Saudi Arabia. The questionnaire included closed and open ended questions that all the respondents were supposed to take. The questions posed within the survey questionnaire included general inquiries on the willingness and competence of the various respondents within the population to participate in the survey. The respondents were persons with experience in matters to do with information systems and e-government operations. The survey also included questions relating to the knowledge and understanding of the various respondents regarding cyberspaces. The questionnaire was taken through an online form with the help of primary researchers who also highlighted the importance of the research to the respondents. Secondary researchers took the research feedback and scrutinized them using different statistical approaches and tools. The questionnaire was quite comprehensive and covered different aspects of the research study.

*3.4. Sample Size*

The sample size for this research was selected at random, with various respondents from the target population being requested to complete the online form. A sample size of 120 respondents with extensive experience in the topic under investigation were engaged for purposes of this research. Twenty of the persons who participated in the research study comprised of persons from fields other than information systems and information technology. All the persons who were engaged for the survey were requested to give consent before responding to the various questions posed within the questionnaire. The sample size taken for this research of 120 respondents was arrived at as a consequence of making different considerations around resources, time constraints, as well as manageability of data by primary and secondary investigators.

## 4. Results and Findings

*4.1. Introduction*

4.1.1. Distribution of Academic and Professional Qualifications

The distribution of the academic and professional qualifications of the persons who participated in the survey exercise were indicative of a very competent population able to meet the requirements of the study. The academic and professions and qualifications of the different respondents who participated in the study is shown in Figure 1 on the distribution of the professions and qualifications of respondents below with approximately 52% being qualified in information technology, about 33% being qualified in government systems, around 7% being qualified in public policy and 8% being qualified in e-government systems. The respondents who signed up for the survey were sufficiently qualified to meet the requirements of the research.

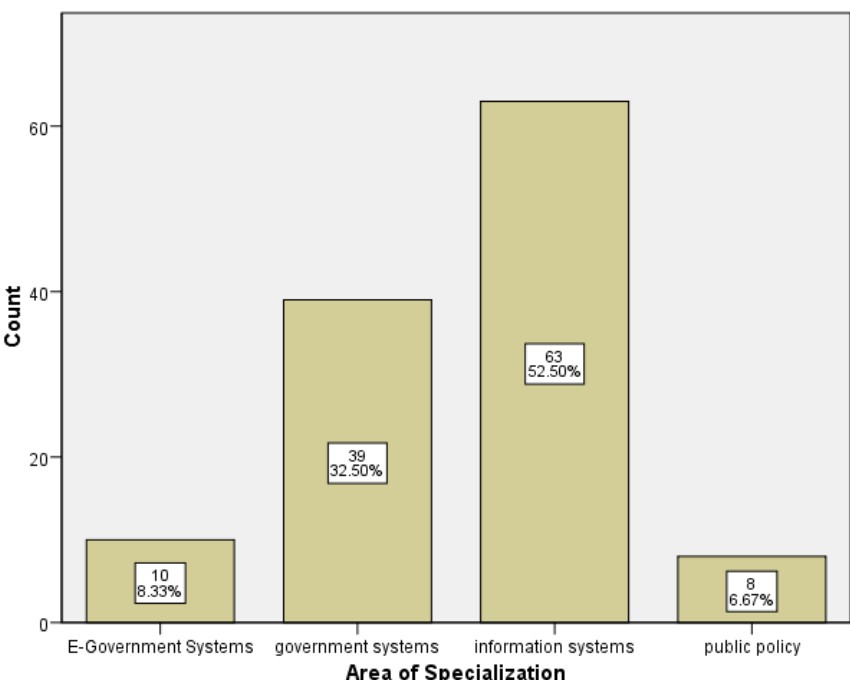

**Figure 1.** Distribution of the proportion of the professions and academic qualifications of the 120 respondents.

4.1.2. Distribution of Years of Experience

The respondents who participated in the research study had varied years of experience as illustrated in Figure 2. A proportion 26% of the population of persons engaged for the research exercise had between 0 to 5 years of experience, approximately 33% of the entire population engaged indicated that they had 6 to 10 years of experience, while

approximately 20% had between 11 to 15 years of experience. Around 21% of the population of respondents interviewed indicated that they had between 16 to 20 years of experience. There were no respondents in the research exercise with years of experience that exceeded 20 years. These results were indicative of a very matured and competent population of respondents regarding the various questions that were posed within the survey.

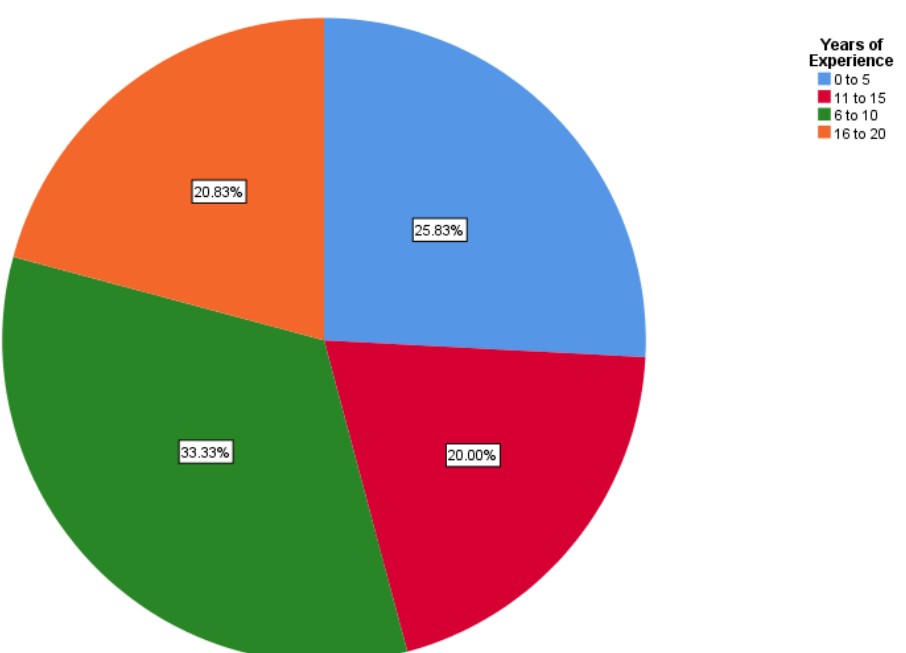

**Figure 2.** Distribution of years of experience among the respondents.

### 4.2. Experimental Analysis

The questions of the research survey were structured to respond to the following primary objectives of the paper:

#### 4.2.1. To Identify Areas of Weakness in the Cybersecurity Framework in the E-Government Operations of Saudi Arabia

A majority of the respondents of the research indicated that they viewed no significant weaknesses in the cybersecurity environment of Saudi Arabia. Most of the respondents indicated that they had confidence in the capabilities of the government of Saudi Arabia to respond to any cybersecurity issues arising within the e-government framework of the country. The respondents further indicated that they believed that the government of the country had taken enough measures to ensure that the country did not face any adversities within the cybersecurity of e-government operations in the long run. This research results are indicative of a relatively safe cybersecurity environment for the e-government infrastructure of Saudi Arabia.

#### 4.2.2. To Establish The Level of Strength in the Cybersecurity Framework of the E-Government Operations of Saudi Arabia

A significant proportion of the respondents of the study indicated that they believed the cybersecurity framework of the country was strong. Most of the respondents intimated that the country had taken sufficient measures to ensure that the country procured some of the most robust systems to safeguard the interests of its citizenry. The respondents indicated that they believed the information of the citizenry of Saudi Arabia was secure and that external threats would not easily compromise the system of service delivery through e-government platforms. The respondents seem to suggest that the cybersecurity environment of e-government in Saudi Arabia is quite robust given the risks that face them in the present.

### 4.2.3. To Compare the Strength of Cyber Security Framework of the E-Government of Saudi Arabia to Other Countries of the World

A majority of respondents who took the survey indicated that the e-government framework of Saudi Arabia compares relatively well with those of other countries of the world. The respondents also indicated that the cybersecurity framework of e-government platforms in Saudi Arabia compares quite competitively with those of major countries of the world. The respondents further noted, however, that the cybersecurity training within different workforce cadres of Saudi Arabia were still very low and needed corrective action from government. The respondents indicated that Saudi Arabia needs to be more proactive in improving the cybersecurity framework from a capacity building point of view.

### 4.3. Analysis of Results and Comparisons

An analysis of the results is indicative of a very bullish outlook on the cybersecurity environment and capabilities of e-government mechanisms in Saudi Arabia. There is an overall optimistic outlook on the safeguards that the country has placed within its information systems from industry experts. The orientation towards improving cybersecurity and e-government operations is primarily driven by the fact that the company has the resources to sustain high quality infrastructure and compete with other countries on the global stage. The country is keen on improving the social welfare of its people, however, and will continually work towards the improvement of infrastructure that is aimed at delivering effective services. The fact that there are pressures related to technological advancements adds to the continuum of improvements observed in the past decade. This generally positive outlook is expected to continue for the next few years as Saudi Arabia aligns to different domestic and international interests.

### 4.4. Validation and Verification

The information inferred from the results of the survey are reflected within the literature review section. There is sufficient evidence to the effect that there is little capacity among locals in Saudi Arabia to deal with issues of technology. The fact that a majority of the people driving the agenda regarding the future of technology in the country are foreigners portend significant risks for the country. The proportion of local persons influencing technological advancements in the country are limited to advisory and oversight services that may lack the technical capacity to deal with issues of cybersecurity. An over-reliance on foreigners to drive different technologies in the country also place the Saudi Arabia at a risk of losing essential intellectual property rights to competing nations and thereby reduce their ability to defend themselves. There is a necessity for the gradual improvement of the cybersecurity framework of Saudi Arabia with the view of giving the government full control of its e-government assets.

## 5. Discussion and Analysis

There is a growing concern regarding security threats to the e-government framework of Saudi Arabia. The country has become increasingly reliant on information technology within different domains. The National Information Security Strategy (NISS), among other entities, has become a significant investment of importance in the context of Saudi Arabia as the country seeks to establish itself in this field. Cyber security is becoming a very big business within Saudi Arabia as both public and private entities are recognizing the growth of the security issues. All major companies are dedicating significant resources towards the effective protection of their information technology infrastructure. There are various issues of importance that the government of Saudi Arabia must continue focusing on as it seeks to strengthen its cybersecurity framework. Some of the areas of concern, in this regard, include creating national guidelines for information security management based on international standards and best practices, developing resilience in information systems; increasing awareness of security risks, and increasing the security and integrity of online information thereby promoting greater use of information technology. The gov-

ernment of Saudi Arabia must also concern itself with the systematic improvement of current infrastructure given the pace at which other countries are developing. These issues form a central area for concern within the context of cyber security issues within the country's e-government framework. Saudi Arabia is a non-industrial nation in the middle east that has witnessed significant socioeconomic development in the past few decades. The implementation of a comprehensive e-government system in Saudi Arabia has been in the works since the early 2000s. This has drawn a large number of multinational IT and security companies to the market, with some of them forming innovative partnerships with local IT and telecommunications firms. An official council has been in force to steer the process and deal with the usage of the venture throughout the period [19]. The people who sit in the advisory group for the implementation of e-government systems in Saudi Arabia have been drawn from various government services and commissions in Saudi Arabia [18]. The advisory group on the implementation of e-government systems has been tasked with re-appropriating the execution of the e-government venture within the country and reporting the same to the Saudi government. Different areas of concern have emerged within the cybersecurity environment of Saudi Arabia over the course of the implementation of various strategies to secure it in the short term and long run. Issues of public activity, protection and general wellbeing of mechanisms around the operationalization of e-government systems has become very essential within in the evaluation of current and future capabilities of the government [19]. The contextual analysis of this paper further reveals that there has been a sense of speed of implementing modernistic approaches towards the implementation of e-government systems in the country with due regard to developments in other countries. Saudi Arabia continues to strengthen its e-government systems by the day as the country seeks to compete at the international stage in terms of service delivery and general satisfaction of the population. Saudi Arabia must now work towards the systematic improvement of its human resource capabilities within its various e-government operations to ensure that it secures its information systems holistically. There seems to be inadequacy in terms of the level of technical training among the people involved in e-government operations across different domains of service delivery in the country [18]. The government of Saudi Arabia must ensure that it actively provides technological related courses among its domestic populace to ensure sustainability of safeguards against cybersecurity attacks. The fact that the country lacks human capital with relevant skills in this area can be inferred by the fact that the general population of the country heavily relies on external labor. Saudi Arabia needs to take active steps to ensure that it reduces the risks associated with inadequacy of capacity to respond to cybersecurity issues within the country's e-government operations. The threats incidental on the cybersecurity framework of e-government systems in Saudi Arabia is vast and complex. The recognition that the Kingdom faces a diversity of growing threats in cybersecurity has necessitated a quick and multi-faceted response to the problems. Global interconnectivity creates significantly adverse and constantly evolving vulnerabilities. These issues also present new types of threatening variables to the Kingdom's cultural and economic activities. Threatening variables are likely to result in various cases of shutdown, corruption or even destruction of communication and information technologies. Threats incidental on the cybersecurity framework of the Kingdom must be managed within appropriate constructs for sustained positive outcomes. The world is moving from over-reliance on the physical exchange and trade of goods and services to a dispensation where information is as significant to holistic development. New technologies have revolutionized the global socioeconomic landscape resulting in numerous developments in the way information is managed and shared in the contemporary informational order [13]. Information and communication systems continue to experience multi-faceted improvements with the advent of new computational strengths as noted by Alfayad & Abbott, (2017). The shift from the industrial age to an informational age is not without challenges, however, necessitating a robust framework for the protection of different interests [25]. The protection, security and safety of online activity is becoming an issue of collective concern within commercial and non-

commercial cyber spaces. The relationship between people, businesses and government is become integrated with improvement in the internet of things (IoT) and technology in general [18]. Cyber security has become an essential issue for consideration in examining the future of technology in supporting societal development. Threats to cybersecurity exploit the complexity and connectivity of essential e-government infrastructure systems thereby placing the security, economy, public safety and health of the people of a given governmental jurisdiction at risk. Governments across the world stress on the issue of cybercrimes due to instances of reputational damage of data breaches in numerous organizations and government systems [26]. Government agencies responsible for critical e-government infrastructure require a consistent and iterative approach for identification, assessment, and management of cybersecurity risks [27]. The case of Saudi Arabia is perfect for consideration, in this regard, due to the variables that characterize its various e-government operations. This analysis provides a comprehensive outlook on the nature of e-government in Saudi Arabia and recommendations for the improvement of cybersecurity in the long run. There is significant literature on the cybersecurity framework of different organizations in Saudi Arabia. There is also extensive literature on the operationalization of e-government systems in Saudi Arabia. Most of these literature fails to place emphasis on the most fundamental and consequential aspect of information systems within any environment in the form of cybersecurity [12]. The question of cybersecurity systems that keep up with current trends in technology must be considered within the new dispensation of digitization of all processes to ensure that information is safeguarded [18]. The general state of technological infrastructure in Saudi Arabia has significantly improved over the past five decades with the growing financial strength of most Middle Eastern countries. Competitiveness within the region and across the world is necessitating a protection of domestic technological interests both within physical and virtual spaces. A CPS is for the most part characterized as a mix of physical (mechanical) segments, transducers (sensors and actuators), and data innovation (IT) frameworks (organization/correspondence frameworks and calculation/examination/control frameworks). A few definitions incorporate the human, like the machine administrator. All in all, a CPS is an actual world framework (machine just or machine in addition to human) that is associated with the digital world. A CPS can be either a shut circle or open-circle framework; implying that it might detect this present reality boundaries of the actual framework and control it, or it might simply detect this present reality boundaries and make these accessible for insightful purposes. IoT or IIoT is by and large characterized as a blend of any of the accompanying: identifiable articles, (for example, RFID labels), information objects (like sensors), intelligent items (like actuators) and keen items, (for example, programming segments that follow up on sensor information for any reason, including pre-preparing, control, examination, and so forth). A Digital Twin is an advanced copy of an actual resource. The meaning of a Digital Twin underscores the association between the physical and the computerized imitation, and the information that is created utilizing sensors. A Digital Twin coordinates transducers, man-made consciousness/AI, information investigation and setting mindfulness. An illustration of setting mindfulness is a savvy indoor regulator, which detects who is available, so the individual's inclinations for encompassing conditions can be contemplated.

The CPS idea arose essentially from a frameworks designing and control point of view, though the IoT idea arose fundamentally from a systems administration and IT viewpoint with beginnings in the RFID setting. The Digital Twin idea then again, arose out of a computerized reasoning/AI viewpoint. Regardless, each of the three can be and are being utilized reciprocally, given that the meanings of the three ideas are combining over the long haul. Diamond Precare specialist innovation IP rides every one of the three definitions and subsequently GEM alludes to CPS, IoT/IIoT and Digital Twin conversely. Jewel specialists procure information on status, activity, surrounding conditions, administrator on the up and up, just as different parts of the activity of a machine, coming about in a multi-dimensional Digital Twin portrayal of a machine. The specialists appoint semantical importance to the information, making an accurate computerized copy of the machine's

noticeable/non-apparent flagging interface. Diamond specialists can be added substance to a current insider savvy regulator, like a PLC, they can be incorporated into the on top of it regulator, or they can incorporate the on top of it regulator. For instance, consider a wafer pick-and-spot machine. The machine gets wafers from a plate and stores it on a transport line prompting another machine for additional preparing. At the point when the plate is unfilled the pick-and-spot machine stops and demands help to top up the plate once more. The plate being vacant is recognized through a photograph sensor and the pick-and-spot instrument utilizes vacuum incitation to get and deliver the wafers. A vacuum sensor detects vacuum pressure. A typical circumstance found is that the photograph sensor and vacuum sensor signals are simply accessible to a coordinated regulator, like a PLC for example. An advanced portrayal of the machine would have to incorporate admittance to these signs. The GEM Precare specialist equipment will detect voltage level changes on the photograph sensor vacuum sensor signal lines. The GEM Precare programming specialist will keep tally of the vacuum sensor's sign line changes to check the quantity of wafers got and kept. Simultaneously the specialist will screen the photograph sensor's sign line to flag when the plate is vacant. The specialist sends the 'plate unfilled' occasion with time stamp to the cloud (private, public or half breed), from where it tends to be accounted for continuously and used to figure MTBA (mean-time-between-assistance). Likewise the GEM specialist will keep check of the quantity of wafers that were gotten and kept, just as start and end time, by observing the vacuum sensor's sign line voltage level changes. The specialist sends this information to FactoryLogix MES, which furnishes the contextualization of the information with other information from related cycles to convey more nitty gritty outcomes and KPIs, like OEE. Establishment and getting specialists to begin streaming information progressively commonly does not need over 24 h and does not need the machine to be halted or shut down. Diamond specialist innovation IP is application-freethinker and can be sent in any mechanical or purchaser application. Likewise, GEM has created market driving topic experience specifically in keen assembling, with center around semiconductor and gadgets fabricating. Precare middleware flawlessly interfaces GEM Precare specialists to the Aegis' FactoryLogix MES stage. This furnishes producers in a real sense for the time being with a move up to shrewd assembling without the expense of redesiging their production line floors with new gear. The joined GEM Precare and Aegis' FactoryLogix arrangement furnishes makers with prescient upkeep, MTBF, MTBA, OEE, accessibility, execution and quality notwithstanding the broad MES highlights accessible in FactoryLogix.

## 6. Conclusions

In conclusion, cybersecurity is one of the most fundamental and essential aspects in the implementation of any e-government system. Cyber security structures must be planned and adapted around the needs of the people that are served by a given e-government system. The case of Saudi Arabia provides essential insight into the most critical areas for consideration in the systematic implementation of a cybersecurity framework in e-governments. Customization instruments within the structure for the implementation of cyber security systems must be driven by a desire to command organizations to achieve results. Adaptability permits the effective utilization of cyber security framework by organizations or agencies, which can portend the beginning of setting up a robust online protection program.

### 6.1. Summary of Findings

A good cybersecurity framework provides a methodical strategy for overseeing network safety hazards within diverse e-government environments. The cyber security framework for e-government in Saudi Arabia is quite comprehensive and incorporates exercises that are fused with diverse network protection programs. These programs can be customized and fitted to address any matters arising within the protection framework. The cybersecurity framework of the e-government infrastructure of Saudi Arabia intends to

supplement and not to supplant network safety programs and cross cutting risks. There are still various areas of improvement including a systematic building of capacity among all work groups in different levels of government operations. The government of Saudi Arabia has done a lot in terms of securing its cyber spaces but needs to make comprehensive considerations for the future.

*6.2. Ethical Consideration*

Significant ethical considerations with far reaching implications have been taken within the research design to ensure that it observes general expectations around the integrity of academic research. The respondents of the survey exercise for this paper were chosen at random and were asked to provide their consent before taking part in the survey. Cautionary measures were taken not to ensure that the rights of persons participating in the survey were not infringed in any way. All the materials and academic articles that were used in the research, including past researchers, were cited accordingly in-text as well as referenced at the end of the paper. This paper took all the necessary measures to ensure that the research study remained valid, reliable and credible academically. No other ethical concerns came to the attention of the research during the planning and research phases of the paper.

**Author Contributions:** Conceptualization, T.A.; methodology, T.A. and A.A.; validation, A.A.; formal analysis, T.A. and A.A.; investigation, T.A. and A.A.; writing—original draft preparation,T.A. and A.A.; writing—review and editing, T.A. All authors have read and agreed to the published version of the manuscript.

**Funding:** This research received no external funding.

**Institutional Review Board Statement:** The study was conducted according to the guidelines of the Declaration of Helsinki, and approved by the Institutional Ethics Committee of Majmaah University (protocol code: MUREC-Dec.25/COM-2019/16-2, date: 25 December 2019).

**Informed Consent Statement:** Informed consent was obtained from all subjects involved in the study.

**Acknowledgments:** The authors would like to thank the Deanship of Scientific Research at Majmaah University for supporting this work.

**Conflicts of Interest:** The authors declare no conflict of interest.

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
