# Peer review of "Developing a Cybersecurity Framework for e-Government Project in the Kingdom of Saudi Arabia"

_jcp, doi:10.3390/jcp1020017_

Round 1

Reviewer 1 Report

This paper provides an in-depth analysis of cybersecurity within the context of information systems. It does an excellent job of promoting the transformation of the E-Government to a secure one. It is a very interesting research in a relevant area. The presentation is satisfactory. This paper could be accepted for publication after a minor revision. The authors may give more discussions in the literature review.

The cybersecurity issues could be categorized into three levels, namely, the process level, the data level, and the infrastructure level. There exist many studies on how to incorporate different blockchain technologies to enhance the security, transparency, and traceability of IT systems. For instance, “blockchain security: a survey of techniques and research directions”. The authors may give more discussions on this issue.

The blockchain-based system could act as an anti-counterfeiting digital twin to ensure that the systems have not been tampered with. The authors may give more discussions on the relationship between the digital twin and the physical system.

In the system/product anti-counterfeiting aspect, a biological feature or edible chemical signature (besides the physical QR, RFID and NFC) may be useful. In the system organization aspect, the blockchain may be combined with optimization models for the complex iteration of several stakeholders. The authors may give more discussions on this issue.

Author Response

We have thoroughly reviewed the comments. We already incorporated the suggestion/comments into the context by enriching the literature review to cover all the required information is under our approach and scope. 

Reviewer 2 Report

The issue of cybersecurity is a timely and important topic that requires interdisciplinary research. Against this background, the peer-reviewed article should be viewed positively, as it concerns the fundamental problem of determining the level of cybersecurity within the e-government framework on the example of the Kingdom Saudi Arabia. The authors also compared the existing cybersecurity solutions with other countries.

The paper contains the methodological foundations of the research, i.e., research goals and research problems. In addition, the peer-reviewed article includes a literature review and the results of empirical research. The manuscript contains conclusions regarding the formulated research problems. The adopted objectives of the article have been achieved.

The structure of the paper and the research methodology are adequate to the subject of the article. The literature has been selected correctly and contains current publications on the research topic.

Comments:

1) In the opinion of the reviewer, it would be worth adding a Digital Object ID (DOI) in the references (if DOI is available).

Author Response

We have already included the DOI as part of the reference
